# Navigate the Unknown: Enhancing LLM Reasoning with Intrinsic Motivation Guided Exploration

## Abstract

Reinforcement Learning (RL) has emerged as a pivotal method for improving the reasoning capabilities of Large Language Models (LLMs). However, prevalent RL approaches such as Proximal Policy Optimization (PPO) and Group Relative Policy Optimization (GRPO) face critical limitations due to their reliance on sparse outcome-based rewards and inadequate mechanisms for incentivizing exploration. These limitations result in inefficient guidance for reasoning. Specifically, sparse rewards fail to deliver sufficient feedback, particularly for challenging problems. Furthermore, such rewards induce systematic biases that prioritize exploitation of familiar trajectories over novel solution discovery. These shortcomings critically hinder performance in complex reasoning tasks, which inherently demand iterative refinement across intermediate steps. To address these challenges, we propose an **I**ntrinsic **M**otivation guid**E**d exploratio**N** me**Th**O**d** fo**R** LLM Reasoning (**i-MENTOR**), a method designed to deliver dense rewards and amplify exploration in the RL-based paradigm. i-MENTOR introduces three innovations: trajectory-aware exploration rewards that mitigate bias in token-level strategies while maintaining computational efficiency; error-conditioned reward allocation to ensure efficient exploration on challenging samples while intrinsically stabilizing training; and advantage-preserving integration that maintains advantage distribution integrity while incorporating exploratory guidance. Experiments across 4 public datasets demonstrate i-MENTOR's effectiveness, achieving a 22.23% improvement on AIME 2024. The source code is available for reference [1].

## 1 Introduction

Reinforcement learning (RL) (Arulkumaran et al., 2017; Shakya et al., 2023; Ladosz et al., 2022) has become essential for training Large Language Models (LLMs) (Hadi et al., 2023; Min et al., 2023), evolving from Proximal Policy Optimization (PPO) (Schulman et al., 2017) to Group Relative Policy Optimization (GRPO) (Shao et al., 2024). Frameworks like DeepSeek (Guo et al., 2025; Liu et al., 2024a; Shao et al., 2024) showcase State-Of-The-Art (SOTA) implementations that optimize multi-step reasoning using trajectory-level advantage estimation. RL uniquely transforms sparse rewards (Gao et al., 2024) into learning gradients, enabling coherent Chain-of-Thought (CoT) (Wei et al., 2022; Plaat et al., 2024) reasoning. By refining action distributions through feedback, RL bridges the gap between linguistic competence and goal-directed problem-solving in LLMs.

While contemporary RL methods like PPO and GRPO demonstrate progress through outcome-based rewards (Uesato et al., 2022; Lightman et al., 2023; Wang et al., 2024a), two fundamental limitations persist for reasoning tasks. First, the reliance on static reward functions creates a sparse learning signal (Bayat et al., 2025; Cao et al., 2024) that fails to guide intermediate reasoning steps due to the limited meaningful rewards on hard samples and early training stages, forcing models to navigate vast action spaces with terminal outcome feedback alone (Bougie & Watanabe, 2025; Song et al., 2025; Bamba et al., 2025). This is particularly evident on difficult samples and datasets (Liu et al., 2024b) where outcome-based rewards are more likely to be zero. Second, despite GRPO's group-wise sampling strategy, the inherent reward structure disincentivizes genuine exploration –

---

[1]https://anonymous.4open.science/r/iMENTOR-C28D

trajectories yielding identical final outcomes receive equivalent advantage estimates regardless of different reasoning CoTs, effectively penalizing computationally intensive sampling efforts (Yu et al., 2025; Liu et al., 2025). This creates a paradoxical scenario where models optimize for reward exploitation at the expense of systematic exploration, particularly detrimental in difficult reasoning tasks that require complicated reasoning processes.

Traditional exploration methods (e.g., RND(Burda et al., 2018), ICM(Pathak et al., 2017), Count-Based Exploration (Ostrovski et al., 2017)) encourage agents to explore novel or under-visited states via intrinsic rewards, inspired by cognitive theories of curiosity. While promising for guiding LLM exploration, these methods face challenges in LLM reasoning tasks due to: **1) Dynamic episodic length and computational overload** arise due to dynamic CoT length and token-level exploration rewards (Burda et al., 2018). Directly using each reasoning step as a sample for traditional exploration methods will result in the exploration reward of the long sequence being higher than that of the short sequence, inducing the model to explore samples with longer sequences. Moreover, the per-token computation for lengthy LLM outputs incurs significant computational overhead. **2) Large action space** poses a challenge due to the exponential growth in possible reasoning paths for LLMs (Zhou et al., 2024; Wen et al., 2024). Naive exploration strategies may incur prohibitive computational costs, negligible outcomes, and even training instability. **3) Exploration reward integration** conflicts with outcome rewards in methods such as GRPO (Shao et al., 2024). The reward normalization in GRPO may invert advantages from positive to negative when exploration bonuses are included, disproportionately penalizing low-exploration samples instead of incentivizing high-value exploration. This introduces destabilizing noise to policy optimization and undermines outcome reward efficacy.

In this paper, we propose an **I**ntrinsic **M**otivation guid**E**d exploratio**N** me**ThO**d fo**R** LLM Reasoning (i-MENTOR), which mitigates reward sparsity, enables smart explorations in LLM training, and effectively resolves the above challenges: **1) Trajectory-aware exploration rewards** operate at the sequence level to eliminate computational overload and sequence-length bias caused by dynamic episodic length. By employing two lightweight trajectory-aware networks, this component efficiently captures reasoning sequence uniqueness while maintaining computational traceability. **2) Error-conditioned reward allocation** selectively applies exploration incentives exclusively to incorrect reasoning trajectories and improves exploration efficiency while maintaining a stable training process. This enables efficient exploration of challenging samples while intrinsically stabilizing training. **3) Advantage-preserving integration** introduces exploration incentives after advantage computation. This approach resolves inherent conflicts between exploration rewards and outcome-based rewards. By preserving the statistical integrity of outcome-driven advantage distributions, it integrates exploratory guidance to enable effective coordination between these objectives. By unifying the above mechanisms, i-MENTOR generates dense, stable, and computationally efficient exploration rewards that seamlessly integrate into RL-based LLM training pipelines. This enables enhanced reasoning capabilities through balanced exploration-exploitation dynamics, overcoming the limitations of conventional exploration rewards in LLM-based sequential decision-making contexts.

Our contributions are summarized as follows: (1) We propose i-MENTOR, a novel method that systematically incorporates dense, stable, and computationally efficient exploration rewards to enhance LLM reasoning. Designed for seamless integration with established RL algorithms, i-MENTOR leverages exploration rewards to improve learning efficiency while maintaining stable policy updates throughout the training process. (2)We present three advances: a) Trajectory-aware rewards generate exploration rewards while maintaining efficiency; b) Error-conditioned reward allocation ensures efficient exploration on challenging samples while intrinsically stabilizing training; c) Advantage-preserving integration injects exploration rewards after advantage computation to preserve policy gradient fidelity. Together, i-MENTOR systematically mitigates reward sparsity and enhances LLM reasoning through coordinated explorations. (3) Experiments on four public datasets and two base LLMs (i.e., Qwen2.5-3B (Team, 2024) and deepseek-7b (DeepSeek-AI, 2025)) demonstrate i-MENTOR's consistent effectiveness across various RL methods and base models, significantly enhancing the reasoning capabilities of LLMs. (4) Our experiments reveal an interesting pattern: i-MENTOR's improvements are more pronounced on difficult datasets. The results suggest that i-MENTOR's dense exploration rewards for each response sequence specifically help models overcome learning barriers in challenging samples. This phenomenon represents a further contribution of our study: it originates from Reinforcement Learning with Verifiable Rewards (RLVR)'s core distinction from traditional RL–leveraging a pre-trained LLM instead of a randomly initialized model. Consequently, exploration yields significantly greater benefits when addressing complex problems.

## 2 METHOD

In this section, we first briefly introduce the current SOTA RL training methods for LLM, i.e., PPO and GRPO, then introduce our method, i-MENTOR.

### 2.1 PRELIMINARY

#### 2.1.1 PROXIMAL POLICY OPTIMIZATION (PPO)

PPO enhances policy optimization through a clipped objective function that ensures stable updates:

$$\mathcal{J}_{\text{PPO}}(\theta) = \mathbb{E}_{(q,a)\sim\mathcal{D}, o_{\leq t}\sim\pi_{\theta_{\text{old}}}(\cdot|q)} \left[ \min\left( w_t(\theta)\hat{A}_t, \text{clip}\left(w_t(\theta), 1-\varepsilon, 1+\varepsilon\right)\hat{A}_t \right) \right] \quad (1)$$

where $w_t(\theta) = \frac{\pi_\theta(o_t|q,o_{<t})}{\pi_{\theta_{\text{old}}}(o_t|q,o_{<t})}$. Here, $(q,a)$ denotes question-answer pairs from dataset $\mathcal{D}$. $o_t$ and $o_{<t}$ are generated responses end at token position $t$ and $t-1$. The current and previous policies are parameterized by $\pi_\theta$ and $\pi_{\theta_{\text{old}}}$ respectively. The advantage estimator $\hat{A}_t$ employs Generalized Advantage Estimation (GAE) (Schulman et al., 2015) with outcome-based reward function $R$ and value function $V$ for advantage estimation, while $\varepsilon$ controls the clipping range. By treating question-response sequences ending at each token position of $o_t$ as distinct states and constraining policy updates within a trust region, PPO stabilizes RL training for LLMs. However, the joint optimization of policy and value functions introduces computational overhead, limiting training efficiency.

#### 2.1.2 GROUP RELATIVE POLICY OPTIMIZATION (GRPO)

GRPO extends PPO through group-wise advantage normalization, formulated as:

$$\mathcal{J}_{\text{GRPO}}(\theta) = \mathbb{E}_{(q,a)\sim\mathcal{D}, \{o_i\}_{i=1}^G \sim \pi_{\theta_{\text{old}}}(\cdot|q)}$$

$$\left[ \frac{1}{G}\sum_{i=1}^{G}\frac{1}{|o_i|}\sum_{t=1}^{|o_i|}\left( \min\left( w_{i,t}(\theta)\hat{A}_i, \text{clip}\left(w_{i,t}(\theta), 1-\varepsilon, 1+\varepsilon\right)\hat{A}_i \right) - \beta D_{\text{KL}}\left(\pi_\theta\|\pi_{\text{ref}}\right) \right) \right]. \quad (2)$$

where $w_{i,t}(\theta) = \frac{\pi_\theta(o_{i,t}|q,o_{i,<t})}{\pi_{\theta_{\text{old}}}(o_{i,t}|q,o_{i,<t})}$, $\hat{A}_i = \frac{R_i - \text{mean}\left(\{R_i\}_{i=1}^G\right)}{\text{std}\left(\{R_i\}_{i=1}^G\right)}$, $G$ responses $\{o_i\}_{i=1}^G$ are generated per input, $\pi_{\text{ref}}$ denotes the reference policy, and $\beta$ controls KL penalty strength. GRPO replaces value function estimation with group-wise normalization on outcome-based rewards, improving training efficiency and effects through reduced computational complexity and group-wise sampling process.

### 2.2 I-MENTOR

Despite advancements, PPO and GRPO face two key limitations: (1) Sparse reward signals that provide limited training guidance, and (2) weak exploration mechanisms during policy updates. These issues hinder learning efficiency and final optimization. To address these challenges, we propose an **I**ntrinsic **M**otivation guid**E**d exploratio**N** me**ThO**d fo**R** LLM Reasoning (i-MENTOR), which introduces structured exploration to enhance LLM reasoning and can be easily implemented in SOTA RL methods like PPO and GRPO. As shown in Figure 1, vanilla RL methods with outcome-based

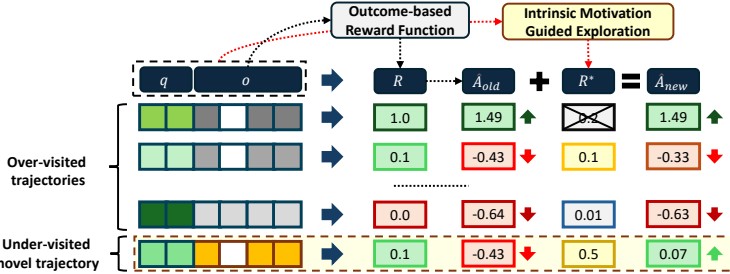

Figure 1: i-MENTOR. Values with green/yellow boxes denote higher outcome-based rewards/advantages and exploration rewards; red/blue denote lower values. Black boxes mark the excluded values. $\hat{A}_{old}$ and $\hat{A}_{new}$ denote advantages derived from outcome-based rewards and i-MENTOR.

> **Question:** You are a helpful assistant. You first think about the reasoning process in the mind and then provide the user with the answer. Using the numbers $[51, 86, 73, 11]$, create an equation that equals 92. You can use basic arithmetic operations $(+, -, *, /)$ and each number can only be used once. Show your work in <think> </think> tags. And return the final answer in <answer> </answer> tags, for example <answer> $(1+2)/3$ </answer>.
>
> **GRPO (Wrong):** <think> First, try $86 + 73 - 51 - 11$; then try $86 + 51 - 73 + 11$; then try $73 + 51 - 86 + 11$; then try $86 + 11 - 73 + 51$; then try $73 + 11 - 86 + 51$; then try $86 + 51 - 11 - 73$; then try $73 + 86 - 51 - 11$; then try $51 + 73 - 86 + 86$ ①; then try $51 + 86 - 73 + 73$ ②; then try $86 + 73 - 51 - 86$; then try $73 + 86 - 11 - 51$; then try $86 + 51 - 73 + 86$; then try $73 + 51 - 86 + 86$; then try $51 + 86 - 73 + 73$ ②; then try $86 + 73 - 11 - 51$; then try $73 + 51 - 86 + 86$. ③
>
> - - - - - - - - - - - - - - - - - - - - - - -
>
> **i-MENTOR-GRPO (Correct):** <think> First, try $86+73-51-11$; then try $86-73+51+11$; then try $73 + 51 - 86 + 11$; then try $51 + 11 + 86 - 73$; then try $86 - 51 + 73 - 11$; then try $73 - 86 + 51 + 11$; then try $51 * 11 - 86 - 73$ ④; then try $(86 - 73) * 11 - 51$. </think> <answer> $(86 - 73) * 11 - 51$ </answer> ③

Figure 2: Case study on Countdown-4 (Gandhi et al., 2024) with simplified responses. Numbers with circles (e.g., ①) highlight key differences. Detailed case study is provided in Appendix A.

rewards focus only on correct trajectories, ignoring under-visited ones and causing LLMs to converge suboptimally. i-MENTOR instead rewards incorrect under-visited trajectories, preventing the model from getting stuck via intrinsic motivation guided exploration.

**A Case Study** is presented in Figure 2 to illustrate the significance of i-MENTOR. Specifically, vanilla GRPO-trained LLM exhibits logical errors (e.g., missing numbers, redundancy in ①), repetitive reasoning patterns (②) due to intensive exploitations without explorations, and failures to solve tasks (③) or apply critical operations like multiplication (④). With the guidance of the exploration rewards from i-MENTOR, LLM trained with i-MENTOR-GRPO reduces logical errors, diversifies reasoning paths (④), and results in a correct answer (③), demonstrating improved reasoning ability. Additionally, as shown in Figure 3(a), i-MENTOR increases the average response length during training, stabilizing at a higher level than vanilla RL methods. This suggests that i-MENTOR 's exploration strategies enable LLMs to learn more complex reasoning processes with longer CoTs through considering diverse reasoning paths in responses, leading to performance gains. In the following subsections, we will systematically introduce i-MENTOR through detailing its key components.

### 2.2.1 TRAJECTORY-AWARE EXPLORATION REWARDS

We propose i-MENTOR to provide intrinsic exploration rewards for LLM reasoning based on the Random Network Distillation (RND) framework (Burda et al., 2018) as it is computationally efficient and does not need prior knowledge.

**RND Network:** The original RND formulation uses a randomly initialized frozen target network $f_{\theta_T}$ and a trainable predictor $f_{\theta_P}$ (same architecture as $f_{\theta_T}$), minimizing prediction error on visited training states:

$$\mathcal{L}(\boldsymbol{s}_t) = \|f_{\theta_P}(\boldsymbol{s}_t) - f_{\theta_T}(\boldsymbol{s}_t)\|_2^2 \tag{3}$$

where $\boldsymbol{s}_t$ denotes the state at step $t$ ($\boldsymbol{s}_t = [q, o_t]$, part of the question-answer sequence for LLM). The predictor updates concurrently with the main RL algorithm. With the same architecture as $f_{\theta_T}$, $f_{\theta_P}$ could converges toward $f_{\theta_T}$ independently of the RL training models. This enables $\mathcal{L}(\boldsymbol{s}_t)$ to serve as an intrinsic reward for exploring novel, high-uncertainty states: a high $\mathcal{L}(\boldsymbol{s}_t)$ indicates poor convergence of $f_{\theta_P}$, reflecting insufficient exploration on similar sequences; conversely, a low loss

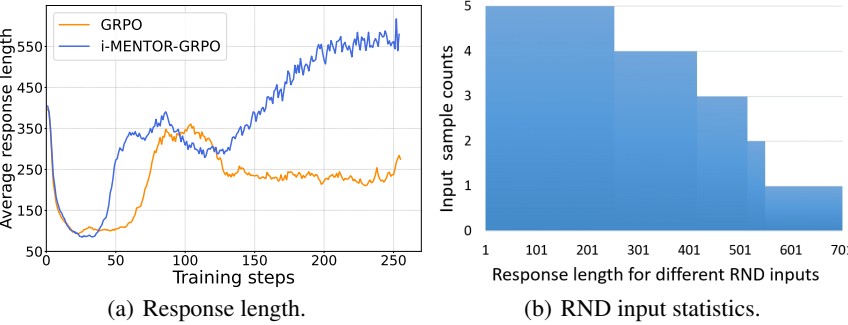

(a) Response length.      (b) RND input statistics.

Figure 3: Training statistics on Countdown-4 (Gandhi et al., 2024).

implies strong familiarity. The exploration reward is consequently defined as:

$$R^{rnd}(\boldsymbol{s}_t) = \frac{\|f_{\theta_P}(\boldsymbol{s}_t) - f_{\theta_T}(\boldsymbol{s}_t)\|_2^2}{\text{std}\left(\|f_{\theta_P}(\boldsymbol{s}_t) - f_{\theta_T}(\boldsymbol{s}_t)\|_2^2\right)}. \tag{4}$$

The denominator uses a moving average of standard deviations during training for a consistent reward scale and temporal stability, but remains fixed during evaluation for determinism.

However, directly applying such a method to LLMs leads to catastrophic performance and efficiency degradation. For LLM reasoning where states correspond to tokens $\boldsymbol{s}_t = [q, o_t]$, direct RND application introduces two challenges: (1) Dynamic episodic length: Sequences $(q, o)$ with dynamic CoT length in response $o$ generate inputs $\{\boldsymbol{s}_t\}_{t=1}^{|o|}$ with lengths $|q| + 1$ to $|q| + |o|$ for RND update. This results in systematic under-sampling of longer input sequences, as they only appear in extended responses, while shorter inputs recur across all sequences regardless of response length, as illustrated in Figure 3(b). (2) Computational overload: Processing thousands of tokens per sequence requires excessive RND network updates.

These phenomena are particularly evident in Figure 3(b). In this figure, a total of $253 + 416 + 515 + 549 + 701 = 2434$ RND inputs are generated from merely 5 rollout responses of 1 question with length $[253, 416, 515, 549, 701]$ in GRPO optimization. Shorter input states dominate these samples, inducing lower exploration rewards from frequent visits. RND requires multiple forward passes per sequence, creating computational overload.

To address these challenges, we propose trajectory-aware exploration rewards operating at the sequence level. Our approach maintains the dual-network architecture but processes complete sequences $(q, o)$ instead of multiple token-level states $\{\boldsymbol{s}_t\}_{t=1}^{|o|}$ as inputs for predictor / target networks:

$$\begin{cases} \mathcal{L}(q, o) = \|f_{\theta_P}(q, o) - f_{\theta_T}(q, o)\|_2^2 \\ R_1^{\star}(q, o) = \dfrac{\|f_{\theta_P}(q, o) - f_{\theta_T}(q, o)\|_2^2}{\text{std}\left(\|f_{\theta_P}(q, o) - f_{\theta_T}(q, o)\|_2^2\right)} \end{cases} \tag{5}$$

Here $\mathcal{L}(q, o)$ is the loss function for updating $f_{\theta_P}$ while $R_1^{\star}(q, o)$ provides a single exploration reward per sequence. This design eliminates token-level bias through uniform sequence treatment and reduces computational complexity from $O(|o|)$ to $O(1)$ per training sequence.

### 2.2.2 ERROR-CONDITIONED REWARD ALLOCATION

Despite the benefits of trajectory-aware exploration rewards, the large action space in LLM reasoning introduces training challenges. Naive exploration strategies become infeasible due to unstable and sparse rewards and suboptimal exploration efficiency. Moreover, uniform exploration rewards across correct and incorrect samples reduce exploration efficiency and induce instability in correct trajectories. To enable more effective exploration, we prioritize samples with residual errors to facilitate diverse discoveries while efficiently improving overall performance. Thus, i-MENTOR employs an error-conditioned reward allocation mechanism across sampling sequences. Specifically, a selection function $I_{a \neq o}$ is designed to allocate exploration rewards exclusively to incorrect samples:

$$R^{\star}(q, o) = I_{a \neq o} \cdot R_1^{\star}(q, o) \tag{6}$$

where $a$ is the ground-truth answer. This conditioning directs exploration resources toward samples with residual errors, enhancing reward utilization while stabilizing predictions for correct samples.

### 2.2.3 ADVANTAGE-PRESERVING INTEGRATION

Since the intensity of the exploration behavior varies at different stages of training, directly combining exploration rewards with outcome-based rewards creates conflicting learning signals across training stages. For PPO, this injects noise into value function estimation. For GRPO, exploration rewards may invert originally positive advantage signs after group normalization, which is a critical issue since exploration rewards should never penalize samples. To resolve this conflict and ensure seamless integration with established RL algorithms like PPO and GRPO, i-MENTOR applies exploration rewards **after** advantage computation and possible value estimation through:

$$\hat{A}_{new} = \hat{A}_{old} + R^{\star}(q, o) \tag{7}$$

where $\hat{A}_{\text{old}}$ denotes the original advantage from outcome-based rewards. For PPO, we apply the exploration reward to the last advantage token. For GRPO, we apply it to each rollout advantage. This design preserves two key properties: (1) The outcome-based advantage maintains its original mean and variance statistics, ensuring stable policy updates; (2) Exploration rewards $R^\star(q, o)$ provide trajectory-level guidance without distorting value estimation. The decoupled formulation prevents exploration rewards from conflicting with outcome-based advantage normalization in GRPO or perturbing value function update in PPO, enabling harmonious integration of both reward components.

By applying trajectory-aware exploration rewards with error-conditioned reward allocation and advantage-preserving integration, i-MENTOR efficiently enhances LLM reasoning by guiding the model toward diverse possible responses during training, while significantly improving overall reasoning ability. The complete algorithm is detailed in Appendix B.

## 3 EXPERIMENTS

Here we evaluate i-MENTOR on three datasets to investigate the following research questions:

- **Q1**: How does i-MENTOR enhance LLM reasoning performance?
- **Q2**: How does i-MENTOR generalize across models and data distributions?

In the following subsections, we begin by outlining the experimental setup, followed by a systematic results analysis addressing our core questions. Training statistics are visualized in Figure 3. Appendix A provides an expanded case study for Figure 2, while implementation specifics–including pseudo code (Appendix B), data statistics (Appendix C), and configuration details (Appendix D)–are deferred for completeness. Beyond core experiments, Appendix E validates i-MENTOR's superiority over basic exploration techniques. We further include an ablation study (Appendix F) quantifying submodule contributions and a sensitivity analysis (Appendix G) examining hyperparameter impacts on performance.

### 3.1 EXPERIMENTAL SETUP

#### 3.1.1 DATASET

We conduct experiments on three public datasets, i.e., GSM8K(Cobbe et al., 2021), Countdown-34 and its harder version Countdown-4(Gandhi et al., 2024) to validate i-MENTOR's effectiveness against vanilla PPO and GRPO algorithms. For computational efficiency, we use a subset of the complete dataset of Countdown-34 and Countdown-4 for training. The detailed dataset statistic is shown in Appendix C. Additionally, we also provide detailed experiments on the AIME 2024 dataset between GRPO and i-MENTOR-GRPO to further validate i-MENTOR's effectiveness with multiple outputs in Table 2.

#### 3.1.2 EVALUATION PROTOCOL

**Environment and LLM backbone:** The experiment environment is built on the TinyZero [2](Pan et al., 2025) and verl [3] framework with Qwen2.5-3B (Yang et al., 2024a; Team, 2024) as base model. We also provide generalization experiments on deepseek-7b in Section 3.4.1. Similar to DAPO (Yu et al., 2025), we exclude the KL penalty from RL algorithms after a detailed analysis in Appendix E.

**Network structure of i-MENTOR:** The predictor and target networks in i-MENTOR share the same architecture, which is dataset-agnostic and LLM-independent. For computational efficiency, we simply apply a lightweight predictor and target network structure for i-MENTOR (embedding size$= 16$, three FFN layers with neurons $[16, 8, 1]$), ensuring negligible additional training time. To make a fairer comparison, we apply a set of fixed training hyperparameters to all datasets.

**Evaluation metric:** For evaluation, we report the average accuracy on five runs for each experiment.

Implementation details can be found in Appendix D to further elaborate on the training settings.

---

[2]https://github.com/Jiayi-Pan/TinyZero
[3]https://github.com/volcengine/verl

Table 1: Average accuracy. i-MENTOR-PPO/GRPO denotes the implementation of i-MENTOR on PPO/GRPO. "Improve" indicates relative improvement.

| Dataset | PPO | i-MENTOR-PPO | **Improve** | GRPO | i-MENTOR-GRPO | **Improve** |
|---|---|---|---|---|---|---|
| GSM8K | 0.8051±0.0057 | 0.8169±0.0042 | **1.47%** | 0.8082±0.0102 | 0.8251±0.0121 | **2.09%** |
| Countdown-34 | 0.5526±0.0332 | 0.5924±0.0273 | **7.2%** | 0.6711±0.0213 | 0.7132±0.0199 | **6.27%** |
| Countdown-4 | 0.3307±0.0242 | 0.3812±0.0281 | **15.27%** | 0.3872±0.0452 | 0.4739±0.0574 | **22.39%** |

## 3.2 MAIN RESULT (Q1)

The main results are presented in Table 1. Figure 4 highlights key training details via some of the training trajectories, showcasing the performance improvements achieved during training.

From Table 1, 2 and Figure 4 we could conclude that:

- GRPO outperforms PPO on all datasets, illustrating superior training effects brought by group-wise sampling and reward normalization. Both i-MENTOR-PPO and i-MENTOR-GRPO outperform standard RL approaches, demonstrating that i-MENTOR effectively guides and enhances the updates of LLM reasoning via exploration behavior during the RL training process. By encouraging the exploration of new responses rather than merely fitting to outcome-based reward through sampling behaviors, i-MENTOR enables LLMs to explore more diverse potential inference paths during training and avoids LLMs getting trapped in local optimal solutions.

- A key challenge in improving LLM reasoning lies in handling difficult samples, which typically yield near-zero rewards that hinder parameter updates. Our experiments reveal an interesting pattern: i-MENTOR's improvements are most significant on Countdown-4, followed by Countdown-34 and GSM8K. This progression aligns with the relative difficulty levels of these datasets, where Countdown-4 > Countdown-34 > GSM8K. The results suggest that i-MENTOR's dense exploration rewards specifically help models overcome learning barriers in challenging samples. This phenomenon represents a further contribution of our study: it originates from RLVR's core distinction from traditional RL–leveraging a pre-trained LLM instead of a randomly initialized model. Consequently, exploration yields significantly greater benefits when addressing complex problems.

- i-MENTOR achieves greater performance gains with GRPO than with PPO. This advantage stems from GRPO's rollout mechanism, which enables i-MENTOR to generate varied exploration rewards for multiple responses to the same question. Such a design promotes broader exploration of potential solution paths compared to single-response PPO updates.

## 3.3 ANALYSIS ON MULTIPLE GENERATIONS (Q1)

In this section, we validate i-MENTOR's efficacy on the AIME 2024 benchmark between GRPO and i-MENTOR-GRPO. For each question, the LLM generates multiple answers to enable more rigorous performance evaluation (Yu et al., 2025). Results in Table 2 demonstrate two key findings: First, LLM accuracy improves consistently with increasing response counts, confirming that aggregating multiple outputs enhances model performance. Second, i-MENTOR achieves approximately

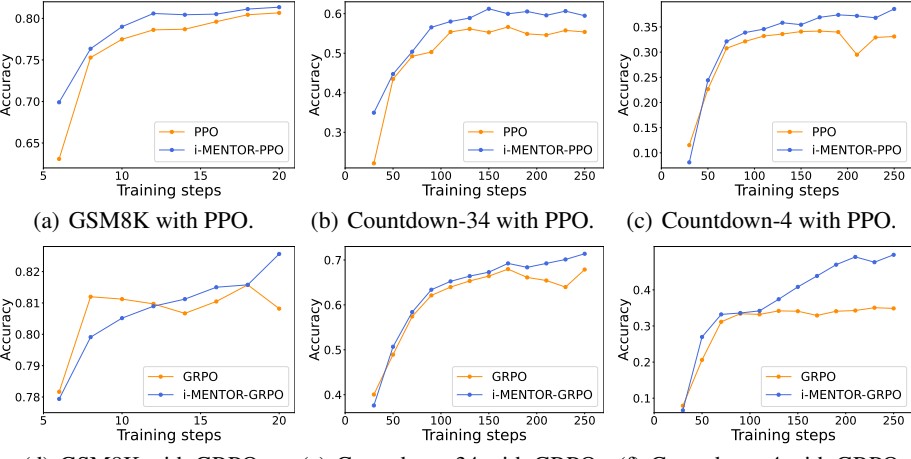

(a) GSM8K with PPO.  (b) Countdown-34 with PPO.  (c) Countdown-4 with PPO.

(d) GSM8K with GRPO.  (e) Countdown-34 with GRPO.  (f) Countdown-4 with GRPO.

Figure 4: Evaluation accuracies in some of the training trajectories.

Table 2: Experimental result on AIME 2024.

| Method | Pass@1 | Pass@8 | Pass@32 | Avg@8 | Avg@32 |
|---|---|---|---|---|---|
| GRPO | 0.3 | 0.3 | 0.3333 | 0.3 | 0.3073 |
| i-MENTOR-GRPO | 0.3667 | 0.3667 | 0.4 | 0.3667 | 0.3667 |
| Improve | 22.23% | 22.23% | 20.01% | 22.23% | 19.33% |

20% improvement across all metrics compared to GRPO baselines. This validates i-MENTOR's effectiveness in solving complex problems while efficiently leveraging multiple responses.

## 3.4 GENERALIZATION ANALYSIS (Q2)

In this section, we conduct experimental analysis to verify the generalization ability of i-MENTOR from both the model and dataset perspectives.

### 3.4.1 BASE MODEL GENERALIZATION ANALYSIS

Here, we verify the effectiveness of i-MENTOR on a new commonly used base model, deepseek-7b, with GRPO on the GSM8K dataset to validate i-MENTOR's generalization on different base models. From the result in Table 3, we could conclude that i-MENTOR can be applied to different base models, including Qwen and DeepSeek, and provide performance gains for their reasoning process.

Table 3: Experimental result with deepseek-7b base model on GSM8K.

| Dataset | GRPO | i-MENTOR-GRPO |
|---|---|---|
| GSM8K | $0.6535\pm0.0049$ | $0.6641\pm0.0052$ |

### 3.4.2 DATASET GENERALIZATION ANALYSIS

This subsection examines i-MENTOR's impact on generalization capability by evaluating performance on out-of-distribution data. Specifically, we train the LLM exclusively on Countdown-4 and measure performance on the unseen Countdown-34 dataset. As evidenced by Table 4, i-MENTOR-GRPO surpasses GRPO on both Countdown-4 and the held-out Countdown-34 dataset after training. This demonstrates that i-MENTOR not only enhances optimization efficacy but also preserves - and even strengthens - the model's generalization capacity when exposed to novel data distributions.

Table 4: Dataset Generalization analysis.

| Dataset | GRPO | iMENTOR-GRPO | **Improve** |
|---|---|---|---|
| Coundown-4 | $0.3831\pm0.0482$ | $0.4726\pm0.0418$ | **23.36%** |
| Countdown-34 | $0.3369\pm0.0531$ | $0.3877\pm0.0563$ | **15.08%** |

## 4 TRAINING TIME ANALYSIS

Here we summarize the training time for experiments in Table 1 to examine the additional cost in terms of time that i-MENTOR incurs. The results in Table 5 demonstrate that i-MENTOR introduces only minimal training time overhead versus baselines without it. This efficiency stems from the predictor and target networks requiring small parameter counts–they comprise several compact linear layers. These networks compute rewards via output differences to determine trajectory visit frequency during training, without needing complex semantic understanding. Critically, since the exploration module operates solely during training, i-MENTOR has zero inference overhead. This confirms i-MENTOR's superiority in delivering performance gains with negligible training time impact.

## 5 EXPLORATION ANALYSIS

In this section, we quantified the effect of GRPO and i-MENTOR-GRPO during the training process to analyze the promoting role of i-MENTOR on the exploration behavior. Specifically, as Section 2.2.1 has stated, the convergence rate of the loss $\mathcal{L}(q, o)$ provided by i-MENTOR represents the frequency of occurrence of similar trajectories during the training process. The higher the frequency of similar trajectories, the faster $\mathcal{L}(q, o)$ decreases on this part of the trajectory, and at the same time, the smaller

Table 5: Training time analysis.

| Dataset | PPO | i-MENTOR-PPO | GRPO | i-MENTOR-GRPO |
|---------|-----|--------------|------|---------------|
| GSM8K | 0.70h | 0.73h | 0.86h | 0.88h |
| Countdown-34 | 7.43h | 7.81h | 12.13h | 14.55h |
| Countdown-4 | 6.97h | 7.20h | 11.69h | 13.03h |

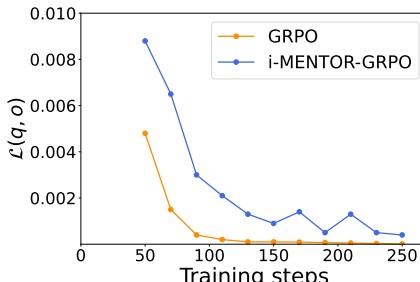

Figure 5: Exploration analysis on Countdown-4.

the average value of $\mathcal{L}(q, o)$ within such batch. Therefore, the decrease speed of this value can be used as a criterion to judge the diversity of trajectories in the training stage.

As shown in Figure 5, the decrease speed of $\mathcal{L}(q, o)$ on GRPO is significantly faster than that of i-MENTOR-GRPO. This proves that the trajectories generated during the training with i-MENTOR-GRPO are more diverse compared to those with GRPO.

## 6 SCALABILITY ANALYSIS

In this section, we explore the scalability of i-MENTOR on different model sizes. Specifically, we experiment the performance of i-MENTOR-GRPO on Countdown-34 with different base model sizes: Qwen2.5- 0.5B/1.5B/3B. The result are shown in Table 6. From the table we could conclude that, i-MENTOR can be applied to models of different sizes, bringing stable performance improvements.

Table 6: Scalability analysis on Countdown-34.

| Model | GRPO | i-MENTOR-GRPO |
|-------|------|---------------|
| Qwen2.5-0.5B | 0.0033 | 0.0107 |
| Qwen2.5-1.5B | 0.4532 | 0.5068 |
| Qwen2.5-3B | 0.6711 | 0.7132 |

## 7 COMPATIBILITY ANALYSIS

In this section, we explore the compatibility of i-MENTOR with other GRPO-based techniques. Specifically, i-MENTOR is lightweight and can be trained alongside any RL method, so it's compatible with improvements like DAPO. Therefore, in this section, we observe the coupling of DAPO and i-MENTOR on Countdown-4 as an exploration of compatibility with i-MENTOR in Table 7. Results in Table 7 show they can work together, boosting performance more than either alone. This further demonstrates the superiority of our method in terms of its compatibility.

Table 7: Compatibility analysis.

| Method | Accuracy |
|--------|----------|
| DAPO | 0.4691 |
| i-MENTOR-DAPO | 0.4924 |

Table 8: Group size analysis.

| Group Size | GRPO | i-MENTOR-GRPO |
|------------|------|---------------|
| 2 | 0.1746 | 0.1909 |
| 3 | 0.3374 | 0.3515 |
| 4 | 0.3593 | 0.4261 |
| 5 | 0.3872 | 0.4739 |

## 8 GROUP SIZE ANALYSIS

Here we observe the influence of different group size in i-MENTOR-GRPO. Specifically, we test i-MENTOR-GRPO with different group size on Countdown-4. From the result in Table 8 we could conclude that, as the group size increases, the performance of i-MENTOR-GRPO steadily improves over GRPO, which proves the compatible effectiveness of group sampling and i-MENTOR.

# 9 RELATED WORKS

## 9.1 REINFORCEMENT LEARNING FOR LLMS

Reinforcement learning has evolved from foundational approaches like PPO (Schulman et al., 2017) to advanced frameworks such as GRPO (Shao et al., 2024). PPO achieves stability in policy updates through clipped objectives and trust region constraints, effectively aligning models for dialogue systems and code generation (Wang et al., 2024b; Shojaee et al., 2023). Though joint policy-value optimization introduces computational overhead (Shao et al., 2024). GRPO (Shao et al., 2024) mitigates these limitations through trajectory-level sampling and group-wise advantage normalization. By generating multiple responses per input and standardizing rewards across trajectory groups, GRPO reduces policy update bias while maintaining efficiency. Recent implementations such as DAPO (Yu et al., 2025) and Dr. GRPO (Liu et al., 2025) further showcase GRPO's scalability through detailed refinement of reasoning objectives, though exploration efficiency remains constrained by static and sparse reward structures (Dou et al., 2025; Gao et al., 2024). Differently, i-MENTOR enhances LLM reasoning through providing dense, stable, and efficient exploration rewards in RL optimization process. These rewards enable intrinsic motivation-guided exploration over diverse response trajectories during training, systematically improving reasoning capabilities.

## 9.2 IMPROVING REASONING ABILITY OF LLMS

Recent advances in LLM reasoning focus on three key paradigms: **Pre-training augmentation** improves foundational reasoning by exposing models to curated datasets and synthetic traces. Models like Qwen (Yang et al., 2024b) and Llama (Grattafiori et al., 2024) show significant gains through domain-specific data scaling, though this requires heavy computational resources and careful data curation. **Prompt engineering** (Sahoo et al., 2024; Chen et al., 2023) enhances reasoning via structured inputs. Techniques like Chain-of-Thought (Yu et al., 2023; Chu et al., 2023) decompose problems into steps, while self-consistency (Wang et al., 2022) improves outputs through majority voting. Extensions such as multi-agent debates (Liang et al., 2023) and iterative refinement (Madaan et al., 2023) further extend capabilities but are limited by manual design and high computational costs. **Algorithmic enhancement** combines inference-time search with RL-based fine-tuning. Methods like Monte Carlo Tree Search (Browne et al., 2012) and RL approaches (e.g., PPO and GRPO) optimize reasoning policies but often overexploit fixed high-reward paths, limiting exploration of novel solutions. Unlike these methods, i-MENTOR actively incentivizes exploration of under-optimized paths through exploration rewards, enabling deeper action-space traversal.

# 10 CONCLUSION

We introduce i-MENTOR, a reinforcement learning framework that systematically addresses the challenge of sparse rewards in LLM reasoning through three methodological advances. First, our trajectory-aware exploration reward employs lightweight networks to provide dense intrinsic rewards while eliminating computational overhead and episodic length bias, enabling efficient discovery of high-quality reasoning paths. Second, we propose error-conditioned reward allocation to ensure efficient exploration on challenging samples while intrinsically stabilizing training in large action spaces, ensuring sustained exploration. Third, our advantage-preserving integration decouples exploration incentives from outcome-based policy gradients, resolving fundamental conflicts between intrinsic and extrinsic rewards. Extensive experiments demonstrate that i-MENTOR significantly enhances reasoning performance when integrated with both PPO and GRPO, establishing its versatility across distinct RL paradigms.

## ETHICS STATEMENT

To the best of our knowledge, there are no ethical issues with this paper.

## REPRODUCIBILITY STATEMENT

We provide our source code at https://anonymous.4open.science/r/iMENTOR-C28D to ensure reproducibility, with detailed implementation details in Appendix D.

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

# A  DETAILED CASE STUDY

We provide the full response texts from the case study (Figure 2) in Figure 6 for reference. Notably, the incorrect responses are longer than the correct ones, a common trend observed across all algorithms examined in this paper. This occurs because a well-trained LLM typically stops generating text once it arrives at the correct answer, whereas incorrect responses involve exploring more possible combinations to find a solution. However, as shown in Figure 3(a), the overall average response length of i-MENTOR-GRPO is longer than that of GRPO. This suggests that LLMs trained with i-MENTOR-GRPO develop more complex reasoning abilities, enabling them to approach problems from a broader range of perspectives.

---

**Question:** You are a helpful assistant. You first think about the reasoning process in the mind and then provide the user with the answer. Using the numbers $[51, 86, 73, 11]$, create an equation that equals 92. You can use basic arithmetic operations $(+, -, *, /)$ and each number can only be used once. Show your work in <think> </think> tags. And return the final answer in <answer> </answer> tags, for example <answer> $(1 + 2)/3$ </answer>.

**GRPO (Wrong):** <think> Let me solve this step by step. First, add 86 and 73 to get 159. Then subtract 51 from 159 to get 108. Finally, subtract 11 from 108 to get 97. However, this does not equal 92. Let's try another combination: add 86 and 51 to get 137. Subtract 73 from 137 to get 64. Finally, add 11 to 64 to get 75. This is still not equal to 92. Let's try another combination: add 73 and 51 to get 124. Subtract 86 from 124 to get 38. Finally, add 11 to 38 to get 49. This is still not equal to 92. Let's try another combination: add 86 and 11 to get 97. Subtract 73 from 97 to get 24. Finally, add 51 to 24 to get 75. This is still not equal to 92. Let's try another combination: add 73 and 11 to get 84. Subtract 86 from 84 to get -2. Finally, add 51 to -2 to get 49. This is still not equal to 92. Let's try another combination: add 86 and 51 to get 137. Subtract 11 from 137 to get 126. Finally, subtract 73 from 126 to get 53. This is still not equal to 92. Let's try another combination: add 73 and 86 to get 159. Subtract 51 from 159 to get 108. Finally, subtract 11 from 108 to get 97. This is still not equal to 92. Let's try another combination: add 51 and 73 to get 124. Subtract 86 from 124 to get 38. Finally, add 86 to 38 to get 124. This is still not equal to 92. Let's try another combination: add 51 and 86 to get 137. Subtract 73 from 137 to get 64. Finally, add 73 to 64 to get 137. This is still not equal to 92. Let's try another combination: add 86 and 73 to get 159. Subtract 51 from 159 to get 108. Finally, subtract 86 from 108 to get 22. This is still not equal to 92. Let's try another combination: add 73 and 86 to get 159. Subtract 11 from 159 to get 148. Finally, subtract 51 from 148 to get 97. This is still not equal to 92. Let's try another combination: add 86 and 51 to get 137. Subtract 73 from 137 to get 64. Finally, add 86 to 64 to get 150. This is still not equal to 92. Let's try another combination: add 73 and 51 to get 124. Subtract 86 from 124 to get 38. Finally, add 86 to 38 to get 124. This is still not equal to 92. Let's try another combination: add 51 and 86 to get 137. Subtract 73 from 137 to get 64. Finally, add 73 to 64 to get 137. This is still not equal to 92. Let's try another combination: add 86 and 73 to get 159. Subtract 11 from 159 to get 148. Finally, subtract 51 from 148 to get 97. This is still not equal to 92. Let's try another combination: add 73 and 51 to get 124. Subtract 86 from 124 to get 38. Finally, add 86 to 38 to get 124. This is still not equal to 92. Let's try another combination: add 51 and 86 to get 137. Subtract (End Here)

- - - - - - - - - - - - - - - - - - - - - - - - - - - - - - - - - - - - - - - -

**i-MENTOR-GRPO (Correct):** <think> Let me solve this step by step. First, calculate 86 + 73 = 159. Then, subtract 51 from the result: 159 - 51 = 108. Finally, subtract 11 from the result: 108 - 11 = 97. However, this does not equal 92. Let's try another combination: calculate 86 - 73 = 13, then add 51: 13 + 51 = 64, and finally add 11: 64 + 11 = 75. This also does not equal 92. Let's try 73 + 51 = 124, then subtract 86: 124 - 86 = 38, and finally add 11: 38 + 11 = 49. This does not equal 92 either. Let's try 51 + 11 = 62, then add 86: 62 + 86 = 148, and finally subtract 73: 148 - 73 = 75. This does not equal 92. Let's try 86 - 51 = 35, add 73: 35 + 73 = 108, and subtract 11: 108 - 11 = 97. This does not equal 92. Let's try 73 - 86 = -13, then add 51: -13 + 51 = 38, and finally add 11: 38 + 11 = 49. This does not equal 92. Let's try 51 * 11 = 561, then subtract 86: 561 - 86 = 475, and finally subtract 73: 475 - 73 = 402. This does not equal 92. Let's try 86 - 73 = 13, then multiply by 11: 13 * 11 = 143, and finally subtract 51: 143 - 51 = 92. </think>
<answer> (86 - 73) * 11 - 51 </answer>

Figure 6: Detailed case study text.

## B   I-MENTOR ALGORITHM

Through the methodological components described in Section 2.2, i-MENTOR implements a reinforcement learning framework with structured exploration guidance to enhance LLM reasoning capabilities. The complete optimization procedure, formalized in Algorithm 1, operates as follows: Given an input batch $B$ of question-response pairs $(q, o)$ for advantage estimation, we first compute the baseline outcome-based reward signals $R$ and initial advantage estimates $\hat{A}_{\text{old}}$ using conventional outcome-based reward functions and RL advantage estimation methods. Subsequently, each sample is processed through i-MENTOR's dual networks $(f_{\theta_P}, f_{\theta_T})$ to simultaneously update the policy network $f_{\theta_P}$ and obtain trajectory-aware exploration rewards $R_1^\star(q, o)$. These exploration rewards are then adaptively scaled through our error-conditioned reward allocation approach, yielding $R^\star(q, o)$ that maintains training stability across different optimization phases. Finally, the refined advantages $\hat{A}_{\text{new}}$ are computed through our advantage-preserving integration mechanism that injects $R^\star(q, o)$ into $\hat{A}_{\text{old}}$ without distorting the original policy gradient signals. The resulting $\hat{A}_{\text{new}}$ subsequently drives policy updates through standard RL optimization algorithms like PPO and GRPO, enabling effective LLM optimization while maintaining gradient stability.

---

**Algorithm 1** Optimization algorithm of i-MENTOR

---

**Input**: A coming batch $B$ of $(q, o)$ samples for advantage estimation. A fixed randomly initialized network $f_{\theta_T}$, a predictor network $f_{\theta_P}$ with identical architecture as $f_{\theta_T}$.

**Output**: Advantage $\hat{A}_{new}$ for policy update with RL algorithms such as PPO and GRPO.

  1: Obtain outcome-based rewards $R$ through outcome-based reward function
  2: Obtain outcome-based advantage $\hat{A}_{old}$ via fixed reward functions
  3: Update $f_{\theta_P}$ according to loss function $\mathcal{L}$ in Equation (5)
  4: Obtain $R_1^\star(q, o)$ via Equation (5)
  5: Obtain $R^\star(q, o)$ via Equation (6)
  6: Add the exploration reward $R^\star(q, o)$ to $\hat{A}_{old}$ for a new advantage $\hat{A}_{new}$ via Equation (7)
  7: return $\hat{A}_{new}$

---

## C   DATASET STATISTICS

This section briefly introduces the datasets used in this paper. Specifically, GSM8K [1] is a dataset of 8.5K high-quality linguistically diverse grade school math word problems. Countdown-34 [2] and Countdown-4 [3] are two mathematical datasets that perform combined operations based on several given numbers to obtain a given answer. Among them, the input sample of Countdown-34 contains 3 or 4 numbers, while the input sample of Countdown-4 only contains four numbers, making its average difficulty higher. For computational efficiency, we use a subset of the complete dataset of Countdown-34 and Countdown-4 for training. The detailed dataset statistic is shown in Table 9.

Table 9: Data Statistics.

| Params | GSM8K | Countdown-34 | Countdown-4 |
|---|---|---|---|
| Training samples | 7,473 | 32,768 | 32,768 |
| Testing samples | 1,319 | 1,024 | 1,024 |
| Max prompt length | 256 | 256 | 256 |
| Max response length | 1,024 | 1,024 | 1,024 |

## D   IMPLEMENTATION DETAILS

In this paper, we use 4 NVIDIA A800 GPUs for each training loop. Due to variations in dataset sizes and difficulty levels, we train for 250 steps on Countdown-3to4 and Countdown-4, and 20 steps on GSM8K, using a batch size of 512 to ensure convergence. Unless otherwise specified, all experiments are conducted with fixed hyperparameters for fair comparison. Based on preliminary grid search

experiments, the exploration intensity parameter $\alpha$ is set to 0.5, and the exploration attenuation rate $\gamma$ is set to 40, ensuring an averaged optimal performance of i-MENTOR across all datasets. For GRPO, the rollout group size is set to 5. For outcome-based rewards, we adopt the same reward function as TinyZero [4], defined as:

$$R(o, a) = \begin{cases} 1.0, & a == o \\ 0.1, & \text{correct format reward} \\ 0.0, & \text{otherwise} \end{cases} \tag{8}$$

where $a$ is the ground truth answer for response $o$. To ensure fair evaluation, we report the average accuracy over five experiments, rather than evaluation scores, to eliminate potential gains from format rewards during evaluation.

## E COMPARISON WITH BASIC EXPLORATION TECHNIQUES

Beyond i-MENTOR, researchers typically control LLM exploration through two basic techniques: (1) adjusting the temperature parameter $Temp$ to influence output diversity by reshaping token probabilities, and (2) modifying the KL penalty coefficient $\beta$ to regulate how strictly the policy adheres to its original behavior during RL updates. We evaluate these approaches using GRPO with varying KL and temperature coefficients, testing whether performance declines when deviating from their optimal values ($\beta = 0.0, Temp = 1.0$) used by default in this paper.

Our experiments in Table 10 reveal that increasing the KL coefficient (which tightens constraints on policy updates) and reducing the temperature (which introduces limited randomness in training) both degrade reasoning performance–the former limits exploratory updates by over-anchoring to the initial policy, while the latter disrupts sample diversity in training. This validates our baseline configuration with $\beta = 0.0$ and $Temp = 1.0$ for all RL methods in this paper. Moreover, by introducing exploration rewards that actively guide the model toward novel reasoning paths, i-MENTOR achieves superior performance. This demonstrates that i-MENTOR 's trajectory-aware exploration rewards complement rather than conflict with basic exploration mechanisms, providing structured guidance for discovering novel responses while maintaining training stability.

Table 10: Comparison with Other Naive Exploration Methods. "$\beta$" and "Temperature" indicate the KL penalty and LLM temperature coefficients.

| Model | Countdown-4 |
|---|---|
| GRPO ($\beta = 0.0, Temp = 1.0$) | 0.3872±0.0452 |
| \w $\beta = 0.001$ | 0.3672±0.0412 |
| \w $\beta = 0.01$ | 0.1778±0.0289 |
| \w $Temp = 0.3$ | 0.2263±0.0272 |
| \w $Temp = 0.6$ | 0.2869±0.0337 |
| i-MENTOR-GRPO ($\beta = 0.0, Temp = 1.0$) | 0.4739±0.0574 |

## F ABLATION STUDY

The results from Table 11 highlight the progressive improvements contributed by each component of our proposed approach, i-MENTOR. Each enhancement effectively addresses core challenges in policy optimization for LLMs, as reflected by the corresponding increase in accuracy at every stage.

Table 11: Ablation study on Countdown-34.

| Model | Accuracy |
|---|---|
| GRPO | 0.6711±0.0213 |
| +Trajectory-aware Exploration Rewards | 0.6939±0.0217 |
| +Error-conditioned Reward Allocation | 0.7065±0.0173 |
| +Advantage-Preserving Reward Integration (i-MENTOR-GRPO) | 0.7132±0.0199 |

From the table, we could conclude that: (1) Incorporating trajectory-aware exploration rewards leads to a notable improvement in reasoning performance. This demonstrates that enabling the model to explore diverse potential responses during training promotes a richer learning process. By encouraging the discovery of alternative reasoning paths rather than converging prematurely on a limited set of high-reward responses, the model acquires more nuanced reasoning capabilities, as evidenced by the marked accuracy gain. (2) Adding the error-conditioned reward allocation mechanism further enhances model performance. By applying error-conditioned reward allocation, i-MENTOR conducts efficient exploration with adaptive balance on exploration and exploitation during policy updates, ensuring stable and effective policy optimization. (3) Finally, the advantage-preserving integration ensures seamless compatibility with RL algorithms, such as PPO and GRPO. By preserving the statistical integrity of outcome-based advantage distributions while incorporating exploratory signals, i-MENTOR achieves a robust balance between exploiting known strategies and exploring new areas, thus enhancing generalization capabilities. With all three components, i-MENTOR-GRPO demonstrates the highest improvement in accuracy.

Overall, the cumulative contributions of these techniques enable i-MENTOR to outperform the baseline GRPO model significantly, achieving a relative accuracy improvement of +6.27%. These results validate the effectiveness of our proposed components in addressing the unique challenges of training LLMs for complex reasoning tasks.

## G   PARAMETER SENSITIVITY ANALYSIS

In addition to Equation (5) and (6), the detailed exploration is obtained with a more specific control on its range and attenuation rate to prioritize early-stage exploration while gradually shifting focus to exploitation, enabling stable policy convergence:

$$
\begin{cases}
r(q,o) = \|f_{\theta_P}(q,o) - f_{\theta_T}(q,o)\|_2^2 \\
R_1^{\star}(q,o) = \alpha \cdot \dfrac{r(q,o) - \min\limits_{(q,o)\in B}(r(q,o))}{\max\limits_{(q,o)\in B}(r(q,o)) - \min\limits_{(q,o)\in B}(r(q,o))} \\
R_2^{\star}(q,o) = I_{a\neq o} \cdot R_1^{\star}(q,o) \\
R^{\star}(q,o) = \dfrac{\gamma}{\gamma + n} \cdot R_2^{\star}(q,o)
\end{cases}
\tag{9}
$$

This process involves two hyperparameters: $\alpha$, which scales the maximum exploration reward intensity, and $\gamma$, which regulates its decay rate. In this section, we visualize the parameter sensitivity experiments on Countdown-4 for i-MENTOR-GRPO in Figure 7 to analyze the influence of both on the performance of i-MENTOR.

Our parameter sensitivity experiments and experiments on other datasets reveal three key insights: (1) Optimal $\alpha$-$\gamma$ combinations vary across datasets due to factors like task complexity. In this paper, $\alpha = 0.5$ and $\gamma = 40$ demonstrate robust performance as averaged defaults across the three datasets. (2) Extreme $\alpha$ values degrade performance–insufficient $\alpha$ limits exploration, while excessive $\alpha$ (especially with value $\gg 1$) risks overemphasizing exploration over correctness (e.g., exploration rewards surpassing the maximum value of outcome-based rewards in some samples). Notably, i-MENTOR with $\alpha = 1.3$ and $\gamma = 100$ still outperforms vanilla GRPO (0.3872), suggesting tolerance to moderate exploration emphasis. (3) Both slow ($\gamma \gg 40$) and rapid ($\gamma \ll 40$) decay rates harm performance: slow decay impedes training convergence, while rapid decay prematurely terminates exploration. This effect amplifies with larger $\alpha$ values.

These findings underscore i-MENTOR's stability within practical parameter ranges while highlighting the necessity of balanced exploration-exploitation dynamics.

## H   LIMITATIONS

While i-MENTOR successfully encourages LLMs to explore novel responses, we believe that incorporating more diverse reward functions to evaluate the multi-faceted value of different responses and enabling multi-angle exploration could further enhance reasoning performance. As a future

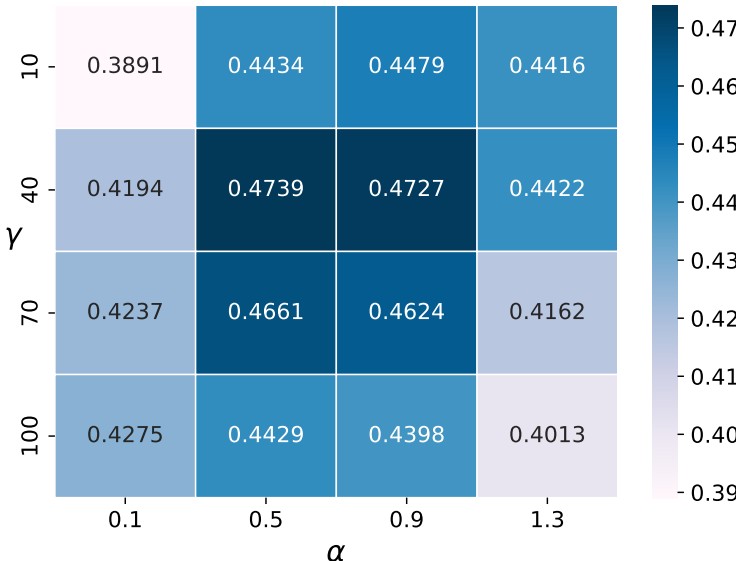

Figure 7: Sensitivity of i-MENTOR to hyperparameters $\alpha$ and $\gamma$ on Countdown-4 for i-MENTOR-GRPO. Optimal performance occurs at $\alpha = 0.5$, $\gamma = 40$, adopted as default settings.

direction, we aim to better analyze these differences to guide LLMs in exploring large action spaces more effectively and developing more complex reasoning capabilities.

