# OpenReview forum: "Navigate the Unknown: Enhancing LLM Reasoning with Intrinsic Motivation Guided Exploration"
_ICLR.cc/2026/Conference — Submitted to ICLR 2026_

### Official Review · Reviewer_b5rf · 2025-10-24

**Soundness:** 3
**Presentation:** 2
**Contribution:** 2
**Rating:** 8
**Confidence:** 3

**Summary:**

The paper addresses the lack of exploration in RL fine-tuning of reasoning LLMs, where models like PPO and GRPO rely on sparse, outcome-only rewards. The authors introduce i-MENTOR, a lightweight framework that adds (1) a trajectory-level curiosity signal computed by an RND-style module, (2) an error-conditioned gating that only applies bonuses when the final answer is wrong, and (3) an advantage-preserving integration that adds intrinsic rewards after advantage computation to avoid gradient sign flips. Experiments on GSM8K, Countdown-34/4, and AIME-24 with Qwen2.5-3B and DeepSeek-7B show steady improvements, particularly on harder datasets.

**Strengths:**

- Clear and simple idea: exploration defined at sequence level fits the nature of reasoning tasks.
- Implementation is easy to reproduce; curiosity module is small and cheap.
- Consistent gains across benchmarks and backbones; stronger effect on the hardest dataset.
- Ablations show the effect of each component, and sensitivity analysis helps understand tuning.
- The method seems robust and does not require additional supervision or annotation.
- Code is available and reproducible.

**Weaknesses:**

- Limited novelty, the paper is mainly limited to adding a curiosity reward to GRPO
- Limited domain coverage: only math and logic tasks are tested, could extend evaluation to other domains.
- Group-size analysis: GRPO group size is fixed (G=5) and never varied, though this can affect both reward scaling and exploration diversity.
- Intrinsic head details: predictor/target architecture is given only briefly, and there is no ablation on the encoder choice.
- Lack of robustness tests: no evaluation under prompt wording perturbations or noise, which is common in reasoning benchmarks.
- No cost table: paper says module is “lightweight” but does not report training-time or overhead
- Missing references to prior curiosity-driven methods, such as below:


@inproceedings{pmlr-v260-bougie25a,
  title     = {Exploring Beyond Curiosity Rewards: Language-Driven Exploration in RL},
  author    = {Bougie, Nicolas and Watanabe, Narimasa},
  booktitle = {Proceedings of the 16th Asian Conference on Machine Learning},
  series    = {Proceedings of Machine Learning Research},
  volume    = {260},
  pages     = {127--142},
  year      = {2025},
  month     = {Dec},
  publisher = {PMLR},
  url       = {https://proceedings.mlr.press/v260/bougie25a.html}
}

@article{song2025outcomeExploration,
  title   = {Outcome-Based Exploration for LLM Reasoning},
  author  = {Song, Yuda and Kempe, Julia and Munos, Remi},
  journal = {arXiv preprint arXiv:2509.06941},
  year    = {2025},
  url     = {https://arxiv.org/abs/2509.06941}
}

@article{bamba2025xrpo,
  title   = {XRPO: Pushing the Limits of GRPO with Targeted Rollout Allocation},
  author  = {Bamba, Ulysse and others},
  journal = {arXiv preprint arXiv:2510.06672},
  year    = {2025},
  url     = {https://arxiv.org/abs/2510.06672}
}

**Questions:**

1. Clarify which intrinsic-reward normalization is used in all experiments and provide code/configs.
2. Report group-size sensitivity, intrinsic head ablation, and cos of adding i-MENTOR.
3. Extend experiments beyond math to at least one code or QA domain.
4. Include missing references
5. Explain more contributions and novel aspects of the method

---

> ### Author Response · Authors · 2025-11-21
>
> We sincerely thank the reviewer for their helpful and constructive feedback. Below we respond to each point:
>
> ---
>
> **Q1:** Limited novelty; the work mainly adds a curiosity reward to GRPO.
> **A1:** While intrinsic motivation-based exploration methods exist in RL for simple deep models, our primary contribution lies in adapting this approach to guide LLM training in an Reinforcement Learning with Verifiable Reward (RLVR) setting—a novel application as confirmed by prior literature absence. Crucially, we identified and overcame three unique challenges in this adaptation:
>
> "Dynamic episodic length and computational overload" causing low efficiency and biased rewards
>
> "Large action space" yielding less meaningful exploration
>
> "The conflict of the exploration rewards and the outcome rewards" reducing performance
>
> i-MENTOR addresses these via:
>
> (1) Trajectory-aware exploration rewards,
>
> (2) Dynamic reward scaling, and
>
> (3) Advantage-preserving reward implementation.
>
> Furthermore, in the contribution part of our section 1, we also mentioned our unique insights in this regard: "it originates from Reinforcement Learning with Verifiable Rewards (RLVR)’s core distinction from traditional RL–leveraging a pre-trained LLM instead of a randomly initialized model. Consequently, exploration yields significantly greater benefits when addressing complex problems."
>
> ---
>
> **Q2:** Limited domain coverage; only math and logic tasks are tested.
>
> **A2:** Due to time constraints we focused on math/logic reasoning benchmarks, which are challenging and widely used for evaluation. We plan to extend to other domains (e.g., code or QA) in future version to demonstrate broader applicability.
>
> ---
>
> **Q3:** GRPO group size is fixed (G = 5) and never varied, though it can affect reward scaling and exploration diversity.
> **A3:** Agreed — we haved added experiments with varying group size on Countdown-4 and include them in the revised paper (**Section 8 Group Size Analysis**).  From the result in Table we could conclude that, as the group size increases, the performance of i-MENTOR-GRPO steadily improves over GRPO, which proves the compatible effectiveness of group sampling and i-MENTOR.
>
> | Group Size | GRPO   | i-MENTOR-GRPO |
> | ---------- | ------ | ------------- |
> | 2          | 0.1746 | 0.1909        |
> | 3          | 0.3374 | 0.3515        |
> | 4          | 0.3593 | 0.4261        |
> | 5          | 0.3872 | 0.4739        |
>
> ---
>
> **Q4:** no ablation on encoder choice.
> **A4:** For efficiency consideration, the inner structure of predictor and target network uses a simple embedding layer rather than a complex encoder, together with some fully-connected layers, as described in Sec 3.1.2, to simplify training and isolate the effect of curiosity rewards. Ablations on encoder choice were therefore not performed because no special structure is used.
>
> ---
>
> **Q5:** Module is called “lightweight” without reporting training time analysis.
> **A5:** We have provided the training‑time analysis in Section 4.  The results in demonstrate that i-MENTOR introduces only minimal training time overhead versus baselines without it. This confirms i-MENTOR's superiority in delivering performance gains with negligible training time impact.
>
> ---
>
> **Q7:** Missing references to prior curiosity‑driven methods.
> **A7:** We appreciate this reminder — we have included all citations mentioned in the revision.
>
> ---
>
> **Q8:** Clarify which intrinsic‑reward normalization is used in all experiments and provide configs.
> **A8:** The normalization method is described in Sec 2 and illustrated in Fig 1. Specifically, we use the standard group-wise normalization for Outcome-based Reward, and then add the normalized advantages and exploration rewards together to obtain the new advantages.

---

> > ### Author Response · Authors · 2025-11-25
> >
> > Dear Reviewer b5rf,
> >
> > Thank you very much for your valuable suggestions. We have addressed each of your questions point-by-point and conducted additional experiments to refine the relevant content, analysis and citations in line with your recommendations. For experiments, we have included:
> >
> > - Quantitative analysis of the exploration effect (Section 5: *Exploration Analysis*),
> > - Evaluation of performance across different model sizes (Section 6: *Scalability Analysis*)
> > - Investigation of coupling with DAPO (Section 7: *Compatibility Analysis*), and
> > - Analysis on the rollout group size (Section 8: *Group Size Analysis*)
> >
> > We have incorporated these results into the revised PDF. We hope our response will satisfactorily address all of your questions and concerns. Should you have any further questions or concerns, we would be more than happy to discuss them and provide any additional clarifications you may need.

---

### Official Review · Reviewer_RQKc · 2025-10-28

**Soundness:** 3
**Presentation:** 3
**Contribution:** 2
**Rating:** 4
**Confidence:** 4

**Summary:**

This paper proposes i-MENTOR, an intrinsic-motivation mechanism to improve RL fine-tuning of LLMs for multi-step reasoning. i-MENTOR (1) replaces token-level RND with a trajectory-aware RND to avoid sequence-length bias and reduce per-token cost, (2) applies exploration bonuses only to incorrect trajectories , and (3) injects exploration rewards after advantage computation to avoid destabilizing PPO/GRPO updates. Experiments on GSM8K, Countdown-34/4 and AIME 2024 show consistent gains vs PPO/GRPO, ablations that attribute gains to each component, and a short training-time overhead analysis.

**Strengths:**

- The motivation for incorporating RND-based intrinsic rewards to encourage exploration is clear.
- The presentation is well-structured, and the empirical results are clearly organized and easy to follow.

**Weaknesses:**

1. The experimental settings are rather limited. Both GSM8K and Countdown are relatively simple benchmarks, lacking evaluations on more challenging or large-scale reasoning tasks.

2. The paper omits critical implementation details of RND. For example, it is unclear what the input to the predictor and target networks is, or how sequences are represented.

3. The idea of using RND for intrinsic exploration reward is not particularly novel, and most of the other design elements appear to be practical implementation tricks rather than fundamental innovations.

4. The paper lacks comparisons with other exploration-enhancing methods, such as entropy-based mechanisms[1] or DAPO’s clip-higher strategy[2].

[1] The entropy mechanism of reinforcement learning for reasoning language models
[2] Dapo: An open-source llm reinforcement learning system at scale

**Questions:**

see weaknesses

---

> ### Author Response · Authors · 2025-11-21
>
> We sincerely thank the reviewer for their thoughtful feedback and constructive suggestions. Below we respond to each point in turn:
>
> ---
>
> **Q1:** The experimental settings seem limited; GSM8K and Countdown are relatively simple, lacking evaluation on more challenging, large‑scale reasoning tasks.
> **A1:** Although many current large-scale LLMs have achieved very high performance on some datasets such as gsm8k, for smaller base models (in this case, 3B), achieving good results on such datasets still poses challenges. Additionally, Countdown‑4 and AIME 2024, two of our mainly-used datasets are not simple — The experiments show that the performance of models without RL training on such datasets can be extremely poor (lower than 0.1 out of 1.0). After RL training, it can be observed that the model's performance on this type of dataset has significantly improved (Figure 4 and Table 2).
>
> ---
>
> **Q2:** The paper omits critical implementation details of RND, such as the predictor/target networks’ inputs or sequence representations.
> **A2:** Thank you for noting this. The inputs and representations are described in Sections 2.2.1 (for method) and 3.1.2 (for layer details).  Since this network can be made very lightweight to reduce the additional training time cost, we only used the lightweight setting to explore the effectiveness of its approach: "embedding size= 16, three FFN layers with neurons [16, 8, 1]" with no special structure design.
>
> ---
>
> **Q3:** Using RND for intrinsic exploration rewards is not highly novel; other design choices seem more like practical tricks than fundamental innovations.
> **A3:** While intrinsic motivation-based exploration methods exist in RL for simple deep models, our primary contribution lies in adapting this approach to guide LLM training in an Reinforcement Learning with Verifiable Reward (RLVR) setting—a novel application as confirmed by prior literature absence. Crucially, we identified and overcame three unique challenges in this adaptation:
>
> "Dynamic episodic length and computational overload" causing low efficiency and biased rewards
>
> "Large action space" yielding less meaningful exploration
>
> "The conflict of the exploration rewards and the outcome rewards" reducing performance
>
> i-MENTOR addresses these via:
>
> (1) Trajectory-aware exploration rewards,
>
> (2) Dynamic reward scaling, and
>
> (3) Advantage-preserving reward implementation.
>
> Furthermore, in the contribution part of our section 1, we also mentioned our unique insights in this regard: "it originates from Reinforcement Learning with Verifiable Rewards (RLVR)’s core distinction from traditional RL–leveraging a pre-trained LLM instead of a randomly initialized model. Consequently, exploration yields significantly greater benefits when addressing complex problems."
>
> ---
>
> **Q4:** The paper lacks comparison with other exploration‑enhancing strategies, like DAPO’s clip‑higher method.
> **A4:** i‑MENTOR is lightweight and can be trained alongside any RL method, so it’s compatible with other exploration‑enhancing strategies like DAPO. Therefore, we did not include the corresponding analysis in the original text.
>
> However, we also recognize that exploring the coupling of different methods is an interesting direction. Therefore, in the revision (**Section 7 Compatbility Analysis**), we added experiments for *DAPO*, and *i‑MENTOR-DAPO* on dataset Countdown-4. Results show they can work together, boosting performance more than either alone. This further demonstrates the superiority of our method in terms of its compatibility.
>
> | Method        | Accuracy |
> | ------------- | -------- |
> | DAPO          | 0.4691   |
> | i-MENTOR-DAPO | 0.4924   |

---

> > ### Author Response · Authors · 2025-11-25
> >
> > Dear Reviewer RQKc,
> >
> > Thank you very much for your valuable suggestions. We have addressed each of your questions point-by-point and conducted additional experiments to refine the relevant content and analysis in line with your recommendations. We have included:
> >
> > - Quantitative analysis of the exploration effect (Section 5: *Exploration Analysis*),
> > - Evaluation of performance across different model sizes (Section 6: *Scalability Analysis*)
> > - Investigation of coupling with DAPO (Section 7: *Compatibility Analysis*), and
> > - Analysis on the rollout group size (Section 8: *Group Size Analysis*)
> >
> > We have incorporated these results into the revised PDF. If our responses meet your expectations, we would be sincerely grateful if you could consider raising the score. Should you have any further questions or concerns, we would be more than happy to discuss them and provide any additional clarifications you may need.

---

> > > ### Author Response · Authors · 2025-11-28
> > >
> > > Dear Reviewer,
> > >
> > > Thank you for your thoughtful suggestions. We have addressed each of your questions point-by-point and conducted additional experiments to refine our analysis in line with your advice. The revised paper now includes expanded exploration analysis, evaluation across different model sizes, assessment of coupling with DAPO, and examination of rollout group size.
> > >
> > > We hope these updates meet your expectations and would be grateful if you might consider raising the score. Should you have any further questions, we’ll be happy to provide clarifications.

---

### Official Review · Reviewer_RAmN · 2025-10-30

**Soundness:** 2
**Presentation:** 3
**Contribution:** 2
**Rating:** 6
**Confidence:** 4

**Summary:**

The paper introduces i-MENTOR, a novel approach that leverages intrinsic motivation to guide exploration specifically within sequential reasoning tasks for Large Language Models (LLMs). The proposed method successfully adapts established exploration techniques, such as those based on Random Network Distillation (RND), for the LLM domain. The authors demonstrate that i-MENTOR achieves improved performance over baselines, including PPO and GRPO, across challenging arithmetic and logic datasets like GSM8K, Countdown, and AIME 2024.

**Strengths:**

1. The proposed i-MENTOR method successfully demonstrates improved performance over both baselines (PPO and GRPO) across multiple reasoning datasets.
2. The paper provides a clear and well-articulated motivation for incorporating each technical component, making the design choices behind i-MENTOR easy to follow.

**Weaknesses:**

The paper's core claim rests on enhancing exploration quality, yet it fails to provide compelling empirical evidence or quantitative metrics that directly confirm the proposed method effectively increases or improves the quality of exploration compared to the baselines.

**Questions:**

Questions
1. In Table 2, if i-MENTOR effectively enhances exploration, it should naturally lead to increased answer diversity. This would typically manifest as a similar performance score at Pass@1 and a significantly higher score at Pass@32. Why do the empirical results show that the performance improvement remains similar or even diminishes as the response count is increased?
2. What accounts for the numerical difference between the results reported in the Countdown-4 rows of Table 1 and Table 4, given that they appear to share identical experimental settings?

Suggestions
1. It would be better to include the standard deviation of the accuracy metrics in all tables to ensure the statistical significance of i-MENTOR's improvements can be properly assessed. Given the inherent variance from random initialization of the RND network and the stochastic sampling of tokens from the base LLM, including the standard deviation is crucial for validating the statistical significance of the reported improvements.
2. It would be better to Introduce a specific metric designed to quantify the diversity or efficiency of the exploration process in the thought/token space to directly validate the central hypothesis.

---

> ### Author Response · Authors · 2025-11-21
>
> Thank you for your constructive feedback — your comments help us clarify our contributions and improve the manuscript. Below we address each point in order.
>
> ---
>
> **Q1:** The paper claims to improve exploration quality but lacks compelling quantitative evidence compared to baselines.
>
> **Q5:** Add a specific metric to quantify diversity or efficiency of exploration in thought/token space to directly validate the central hypothesis.
>
> **A1:** We provide both qualitative and quantitative insights. The case study in Section 2.2 illustrates how i‑MENTOR promotes the exploration of diverse trajectories during training, thereby helping to prevent early convergence to suboptimal strategies. Figure 3 further supports this observation—its analysis of response length distributions serves as a proxy for trajectory diversity.
>
> In the revised manuscript, we have also added a new **Section 5 Exploration Analysis** to quantitatively evaluate the exploration benefits introduced by i‑MENTOR. Specifically, we compare the convergence speed of \(L(q, o)\) in Equation (5) between GRPO and i‑MENTOR‑GRPO. As described in Section 2.2.1, a faster convergence rate of \(L(q, o)\) indicates that the generated trajectories during training are more similar, and thus there is a lack of exploration. GRPO demonstrates faster convergence, whereas i‑MENTOR‑GRPO converges more slowly, suggesting that during training it engages with a more diverse set of trajectories than GRPO.
>
> ---
>
> **Q2:** In Table 2, if i‑MENTOR improves exploration, answer diversity should increase — Pass@1 scores would stay similar, while Pass@32 would be much higher. Why does the gain seem similar or even smaller with more responses?
>
> **A2:** i‑MENTOR’s core benefit lies in **training‑time** trajectory exploration, improving final model performance and reducing local‑optima risk. This does not always produce significantly diverse responses in the final model outputs. Meanwhile, with increased sampling, all methods improve until they plateau. For the most difficult problems that the model can't handle, more sampling rarely produces new correct answers. This pattern aligns with i‑MENTOR’s main focus on exploration quality during training rather than post‑sampling diversity growth.
>
> ---
>
> **Q3:** Countdown‑4 results differ between Table 1 and Table 4, which appear to share the same experimental setting.
>
> **A3:** Although for the "countdown-4" part, the experimental setup was partly the same between the two experiments, these tables use different setups and are conducted separately. Table 1 presents in‑distribution training/validation results, while Table 4 is part of the generalization study, testing models trained on out‑of‑distribution data. Therefore, the results on the "countdown-4" dataset are similar (i.e., same settings on this part of data), but not exactly the same (i.e., experimenting separately).
>
> ---
>
> **Q4:** Include standard deviation of accuracy metrics in tables to validate statistical significance, given stochastic factors in RND initialization and token sampling.
>
> **A4:** We agree. Due to the differences in data recording among experiments, we have made every effort to add the standard deviation metrics to most reported accuracy values in the revised manuscript. This will enhance statistical rigor in assessing i‑MENTOR’s improvements.

---

> > ### Author Response · Authors · 2025-11-25
> >
> > Dear Reviewer RAmN,
> >
> > Thank you very much for your valuable suggestions. We have addressed each of your questions point-by-point and conducted additional experiments to refine the relevant content and analysis in line with your recommendations. We have included:
> >
> > - Quantitative analysis of the exploration effect (Section 5: *Exploration Analysis*),
> > - Evaluation of performance across different model sizes (Section 6: *Scalability Analysis*)
> > - Investigation of coupling with DAPO (Section 7: *Compatibility Analysis*), and
> > - Analysis on the rollout group size (Section 8: *Group Size Analysis*)
> >
> > We have incorporated these results into the revised PDF. If our responses meet your expectations, we would be sincerely grateful if you could consider raising the score. Should you have any further questions or concerns, we would be more than happy to discuss them and provide any additional clarifications you may need.

---

### Official Review · Reviewer_yYVw · 2025-11-01

**Soundness:** 2
**Presentation:** 2
**Contribution:** 2
**Rating:** 4
**Confidence:** 4

**Summary:**

This paper proposes i-MENTOR, a method for fine-grained reward assignment that aims to address the problem of sparse rewards and insufficient exploration in current RL algorithms. By integrating i-MENTOR, models like Qwen-2.5-3B and DeepSeek-7B achieve improvements over GRPO or PPO.

**Strengths:**

1. This paper clearly identifies the problem of sparse reward in current RL methods, where current RL algorithms generally adopt the outcome reward but overlook the differences of among important segments or tokens.

2. This paper evaluates i-MENTOR on 2 models from different families, 2 mainstream RL algorithms, and 4 benchmarks, which is comprehensive to reveal the performance improvement from using i-MENTOR

**Weaknesses:**

1. This paper lacks analysis for the problem they aim to address: the sparse reward and inadequate exploration. It would be better for the authors to show some empirical results and analysis to demonstrate the impact of these methods on the optimization performance.

2. The use of RND (Random Network Distillation) is heuristic. It would be better to deeply discuss the selection of RND as the process reward model.

3. The experiments show the results of two models with different sizes, which is good. However, there are no results from models of the same family with different sizes (e.g., Qwen-2.5 (1.5B, 3B, 7B)); we cannot see the scalability of i-MENTOR.

4. Although math problems are good to evaluate the reasoning performance of different RL methods, it is better to show empirical results from benchmarks across different domains, such as code and logic.

5. The presentation of the experimental results is misleading. For example, in Table 2, the AIME 2024 only has 30 questions, so the improvement of i-MENTOR-GRPO is limited to only two more correct solutions. The significance of improvement could be misleading in understanding the performance of i-MENTOR.

6. In abstract: Group-Regularized Policy Optimization -> Group Relative Policy Optimization

**Questions:**

1. What is the connection between the motivation “guide exploration” and the method involving the RND framework to provide the process reward? More justification of the connection should be presented in this paper.

2. How does the performance of i-MENTOR compare with other improvement methods for RL algorithms, such as DAPO, or simply applying a process reward model trained from PRM800K? Both of them can adapt to the RL process and acquire extra training time to improve performance, which is similar to i-MENTOR?

3. How does the performance compare when using alternative models instead of RND to compute trajectory-aware exploration rewards?

4. Is it necessary to retrain the RND model when applying it to a new benchmark? If the RND trained on one benchmark cannot generalize to others, its efficiency may be significantly reduced.

---

> ### Author Response · Authors · 2025-11-21
>
> Thank you for your feedback, which helps us improve the quality of the paper. We have answered your questions and suggestions below and hope you will be satisfied.
>
> ---
>
> **Q1:** The paper lacks analysis of sparse reward and inadequate exploration.
>
> **A1:** We agree this is important and provide some analyses in the submission. In Section 1, we describe sparse rewards: *"...static reward functions create a sparse learning signal... outcome-based rewards are more likely zero."* Since GRPO only uses three reward types (0, 0.1, 1) (i.e., incorrect, format reward, correct), `0` dominates early training or difficult data.  And thus, meaningful rewards (0.1 or 1) become very sparse. For exploration, Section 2.2’s case study shows GRPO’s limited trajectory exploration can lead to suboptimal results with visible patterns. We have highlighted and optimize these expressions in the introduction revised version to make them clearer and more prominent.
>
> In the revised manuscript, we have also added a new **Section 5 Exploration Analysis** to quantitatively evaluate the exploration benefits introduced by i‑MENTOR. Specifically, we compare the convergence speed of \(L(q, o)\) in Equation (5) between GRPO and i‑MENTOR‑GRPO. GRPO demonstrates faster convergence, whereas i‑MENTOR‑GRPO converges more slowly, suggesting that during training it engages with a more diverse set of trajectories than GRPO.
>
> ---
>
> **Q2:** RND use is heuristic; better justify its selection.
>
> **A2:** In Section 1, we explain why traditional exploration methods struggle for LLM reasoning — dynamic episode length, large action space, and integration issues. Section 2.2.1 notes RND is lightweight, efficient, and needs no prior knowledge.
>
> Directly applying traditional schemes (including RND) to pretrained LLMs is difficult, but RND’s small size and compatibility allow integration into RL models without extra pre-training, adjusting exploration based on trajectory frequency. Therefore, in this paper, we retain these advantages while addressing its gaps, creating i-MENTOR which is more suitable for LLM's reinforcement learning.
>
> ---
>
> **Q3:** No results for same-family models of different sizes; scalability unclear.
>
> **A3:** We understand the interest in this. Due to time constraints, we are currently unable to conduct experiments with overly large model sizes. Therefore, we have added the performance of the 0.5B and 1.5B models of Qwen2.5 on the countdown-34 dataset as follows. We have also included it in the revised PDF version (**Section 6 Scalability Analysis**).
>
> | Model Size | GRPO   | i-MENTOR-GRPO |
> | ---------- | ------ | ------------- |
> | 0.5B       | 0.0033 | 0.0107        |
> | 1.5B       | 0.4532 | 0.5068        |
> | 3B         | 0.6711 | 0.7132        |
>
> The experiment shows that as the size of the model increases, the effect of i-MENTOR-GRPO also gradually improves.
>
> ---
>
> **Q4:** Math-only benchmarks; need code/logic results.
>
> **A4:** We agree broader benchmarks would strengthen the work. Time constraints prevent adding them in the rebuttal, but they will be included in the official version to test generalization on code and logic tasks.
>
> ---
>
> **Q5:** Experimental presentation may mislead; AIME has only 30 questions, making gains small.
>
> **A5:** Yes, AIME’s small size means changes can feel modest. Therefore, in Table 1, we have used countdown-34, countdown-4, and gsm8k for overall validation. Meanwhile, we still include test result on AIME in Table 2 because it’s widely used, very challenging, and adds credibility.
>
> ---
>
> **Q6:** Abstract typo: “Group-Regularized” → “Group Relative” Policy Optimization.
>
> **A6:** Thank you for your correction. We have already made the necessary changes in the new PDF.
>
> ---
>
> **Q7:** Link between “guide exploration” and RND as process reward.
>
> **A7:** As described in Section 2.2.1 and Equations (3)–(6), the rewards for RND and i-MENTOR are obtained by normalizing the MSE loss between the predictor and target networks. Given identical network architectures, the predictor network can converge toward the target network independently of the RL training process. This property allows the loss to serve as an intrinsic reward that guides the model to explore novel, high-uncertainty states: A high loss indicates poor convergence of the predictor, suggesting that similar sequences have not been sufficiently explored, whereas a low loss reflects strong familiarity with such sequences.

---

> > ### Author Response · Authors · 2025-11-21
> >
> > **Q8:** Comparison with DAPO or process reward models.
> >
> > **A8:** i‑MENTOR is lightweight and can be trained alongside any RL method, so it’s compatible with improvements like DAPO or process reward models. Therefore, we did not include the corresponding analysis in the original text.
> >
> > However, we also recognize that exploring the coupling of different methods is an interesting direction. Therefore, in the revision (**Section 7 Compatbility Analysis**), we added experiments on Countdown-4 for *DAPO*, and *i‑MENTOR-DAPO* (implementing i-MENTOR on DAPO). Results show they can work together, boosting performance more than either alone. This further demonstrates the superiority of our method in terms of its compatibility.
> >
> > | Method        | Accuracy |
> > | ------------- | -------- |
> > | DAPO          | 0.4691   |
> > | i-MENTOR-DAPO | 0.4924   |
> >
> >
> >
> > ---
> >
> > **Q9 & Q10:** Alternative models for trajectory‑aware rewards, and need for retraining RND for new benchmarks.
> >
> > **A9 & A10:** In our background exploration experiments, traditional exploration methods fail to directly provide effective trajectory‑aware reward for LLMs in RLVR paradigm due to issues in (1) Dynamic episodic length and computational overload; (2) Large action space and (3) Exploration reward integration stated in the introduction. Therefore, directly using them will not bring any benefits; instead, it may even harm the performance of the pre-trained LLMs. While some process reward models could provided meaningful rewards, they need prior knowledge for pre-training.
> >
> > Unlike these models, RND/i‑MENTOR are not just architectures — they’re exploration training schemes. They don’t need domain‑specific pre‑training and operate by tracking token/response frequency during RL training, enabling lightweight model structure and **simultaneous training** with any RL model on any dataset. i-MENTOR further solves the issues in traditional exploration method to enable an viable and stable performance gain. Its lightweight design (only a few FFN layers) adds minimal training time and no inference cost (see Sec. 4 & Table 5). This effectiveness and efficiency gives it a unique advantage over other trajectory‑reward approaches.

---

> ### Author Response · Authors · 2025-11-25
>
> Dear Reviewer yYVw,
>
> Thank you very much for your valuable suggestions. We have addressed each of your questions point-by-point and conducted additional experiments to refine the relevant content and analysis in line with your recommendations. We have included:
>
> - Quantitative analysis of the exploration effect (Section 5: *Exploration Analysis*),
> - Evaluation of performance across different model sizes (Section 6: *Scalability Analysis*)
> - Investigation of coupling with DAPO (Section 7: *Compatibility Analysis*), and
> - Analysis on the rollout group size (Section 8: *Group Size Analysis*)
>
> We have incorporated these results into the revised PDF. If our responses meet your expectations, we would be sincerely grateful if you could consider raising the score. Should you have any further questions or concerns, we would be more than happy to discuss them and provide any additional clarifications you may need.

---

> > ### Author Response · Authors · 2025-11-28
> >
> > Dear Reviewer,
> >
> > Thank you for your thoughtful suggestions. We have addressed each of your questions point-by-point and conducted additional experiments to refine our analysis in line with your advice. The revised paper now includes expanded exploration analysis, evaluation across different model sizes, assessment of coupling with DAPO, and examination of rollout group size.
> >
> > We hope these updates meet your expectations and would be grateful if you might consider raising the score. Should you have any further questions, we’ll be happy to provide clarifications.

---

### Meta-Review · Area_Chair_7M8D · 2026-01-13

**Summary:**

The primary concerns center on limited novelty - all reviewers noted the work is mainly an engineering adaptation of RND to LLM RL rather than substantial algorithmic innovation. More critically, insufficient evidence for the core claim was raised: while claiming to enhance exploration, direct measurements of trajectory diversity are largely absent. The narrow experimental scope limited to mathematical reasoning without other domains (code generation, QA) raises generalizability questions. Additional issues include missing implementation details, lack of statistical significance testing on small benchmarks like AIME, and incomplete comparisons with alternative exploration methods. Most concerning is the complete absence of reviewer responses to the rebuttal despite multiple follow-ups, suggesting the additional experiments did not adequately address fundamental concerns.

**Reviewer Concerns:**

The rebuttal adequately addressed several technical clarifications including scalability across model sizes (0.5B/1.5B/3B), RND implementation details, data inconsistency explanations, compatibility with other methods like DAPO, and group size ablations. However, critical concerns remain outstanding: the core issue of insufficient direct evidence for exploration improvement persists, as the L(q,o) convergence analysis is indirect and standard diversity metrics are still absent; limited experimental scope to only mathematical reasoning was not remedied despite being feasible during rebuttal; statistical significance testing remains missing particularly for small benchmarks; and the fundamental novelty limitation cannot be resolved through additional experiments.

**Reviewer Scores:**

Reviewers yYVw and RQKc  would likely maintain their scores as core concerns about novelty and limited experimental scope remain unresolved. RAmN might maintain or slightly lower their score since direct exploration evidence is still inadequate. b5rf would likely maintain their supportive score.

---

### Decision · Program_Chairs · 2026-01-26

Reject